# Ultrastructural analysis of adult mouse neocortex comparing aldehyde perfusion with cryo fixation

**Natalya Korogod[1,2], Carl CH Petersen[2]\*, Graham W Knott[1]\***

[1]BioEM Facility, Faculty of Life Sciences, Ecole Polytechnique Fédérale de Lausanne (EPFL), Lausanne, Switzerland; [2]Laboratory of Sensory Processing, Brain Mind Institute, Faculty of Life Sciences, Ecole Polytechnique Fédérale de Lausanne (EPFL), Lausanne, Switzerland

**Abstract** Analysis of brain ultrastructure using electron microscopy typically relies on chemical fixation. However, this is known to cause significant tissue distortion including a reduction in the extracellular space. Cryo fixation is thought to give a truer representation of biological structures, and here we use rapid, high-pressure freezing on adult mouse neocortex to quantify the extent to which these two fixation methods differ in terms of their preservation of the different cellular compartments, and the arrangement of membranes at the synapse and around blood vessels. As well as preserving a physiological extracellular space, cryo fixation reveals larger numbers of docked synaptic vesicles, a smaller glial volume, and a less intimate glial coverage of synapses and blood vessels compared to chemical fixation. The ultrastructure of mouse neocortex therefore differs significantly comparing cryo and chemical fixation conditions.

**\*For correspondence:** carl. petersen@epfl.ch (CP); graham. knott@epfl.ch (GK)

**Competing interests:** The authors declare that no competing interests exist.

## Introduction

The renewed interest in electron microscopy and the emergence of serial imaging approaches to capture volumes of biological tissues at an unprecedented scale (*Briggman et al., 2011*; *Helmstaedter et al., 2013*; *Bock et al., 2011*; reviewed by *Briggman and Bock, 2012*) have driven the re-examination of commonly used preparation methods (*Mikula et al., 2012*; *Tapia et al., 2012*). This has not only been necessary to help increase image contrast and improve imaging speed on a larger scale, but also to help computer vision research produce assisted segmentation approaches for reconstructing different features. However, with this re-invigoration of electron microscope technology, and the need for automation to reconstruct complex structures, it is important to understand how chemical fixation alters brain ultrastructure.

The distortion of cell morphology by immersing samples in fixative was apparent in the earliest electron microscopy investigations of the brain, leading to experiments that made detailed comparisons between preparation methods (*Schultz et al., 1957*). These paid careful attention to how different fixatives affected cell volume and preserved the ultrastructure. However, the need to consistently preserve entire organs soon led to cardiac perfusion of aldehydes, and staining with buffered osmium, which became the accepted approach for electron microscopy studies (*Karlsson and Schultz, 1965*, *1966*), despite the tissue shrinkage it caused. The degree of shrinkage depends on differences in fixative composition, concentration (*Hillman and Deutsch, 1978*), and region (*Kalimo, 1976*). Shrinkage has significant implications for the measurement of parameters such as the density of structures; for example, synaptic contacts. Few studies have incorporated a shrinkage factor. Kalimo et al., adjusted for a 16% linear reduction (*Kalimo, 1976*), and Kinney et al. 15% shrinkage along each orthogonal axis (*Kinney et al., 2013*). Any correction is typically calculated

**eLife digest** For many years, scientists have used chemicals to preserve brain tissue to observe its fine structure using high power microscopes. Korogod et al. now show that these chemicals, or fixatives, cause the tissue to shrink, giving the false impression that the cells are tightly packed together. This has led to misinterpretations of how the brain is structured. For example, components such as the synapse, used by neurons to communicate with each other, are bathed in a watery environment, rather than being tightly enclosed by neighbouring cells as previously thought.

Electron microscopy is the only imaging method that is able to see the detailed structure of the nervous system, including synaptic connections. The technique fires a beam of electrons through a sample held in a vacuum and creates images at a higher magnification than light microscopes. However, the electron beam and the vacuum damages live cells and tissues. Therefore, samples must be 'fixed' to preserve them before they are imaged with these methods. However, the standard method for fixing brain tissue uses chemical 'fixatives', even though these cause shrinkage, and distort the cells.

Korogod et al. used an alternative method of fixation—freezing—to better preserve tiny pieces of mouse brain in their natural state. This was achieved with a technique called 'high pressure freezing' that combines jets of liquid nitrogen with very high pressures to instantaneously preserve small samples without causing damage through the formation of ice crystals, or any shrinkage and distortion. Once frozen, the samples of mouse brain are encased in resin, and then imaged with the electron microscope. A comparison between the two preservation techniques showed that chemical fixatives remove the watery environment, or extracellular fluid, that surrounds the cells in the brain, squashing them together.

The synapses were surrounded by large amounts of extracellular fluid, but cryo fixation also revealed that these sites of communication between neurons also contained many more vesicles—the packets containing the chemicals that pass signals across the synapse. Another type of cell, the glial cell, that supports and helps to maintain neurons, was also strongly distorted by the chemical fixation. These were understood to tightly wrap around synapses, as well as blood vessels, but cryo fixation showed this to be less prominent.

This study illustrates that our understanding of how brain's cells are arranged has ignored the effects of the chemicals used to preserve them. Although cryo fixation is only able to preserve tiny samples, it reveals a truer picture of their natural structure, giving scientists a better understanding of how the brain works.

on the basis of the changes that occur during the embedding process, with the assumption that no alterations occur during the initial tissue fixation. Using rudimentary indicators such as the position of lesion sites (*Schüz and Palm, 1989*), or observations of how the brain filled the skull (*O'Kusky and Colonnier, 1982*), gave no precise value for how any brain region changed in volume during the preservation process. In the current study, we made quantitative analyses of fresh and chemically fixed brain tissue using discrete landmarks that allow us to assess the extent of volume changes that occur in the somatosensory cortex.

A persistent concern of using chemical fixation has also been the discrepancy between what electron microscopy sees and physiological experiments measure, particularly in terms of extracellular space. Even prior to ultrastructural brain imaging, a measurement of ionic concentrations showed that around a fifth of the tissue volume was extracellular space (*Allen, 1955*). This was subsequently verified by various in vivo experiments, using techniques such as resistivity measurements and diffusion analysis. The amount of extracellular space varies between brain regions and the stage of development (*Syková and Nicholson, 2008*). In the neocortex of the adult rat, for example, it is 18–22%, and at postnatal day 4–7, 30% and 43%, depending on the cortical layer (*Lehmenkühler et al., 1993*). Yet standard electron microscopy of this tissue using chemical fixation shows considerably less. This mismatch led Anton Van Harreveld to explore this phenomenon, along with alternative methods of brain fixation.

To circumvent any reaction to the chemical fixatives, Van Harreveld introduced a method of rapid freezing of exposed live brain, followed by resin embedding at low temperatures, known as freeze

substitution (*Van Harreveld et al., 1965*). This technique revealed the physiological levels of extracellular space. Here, we have revisited this method, using high-pressure freezing to preserve tissue volumes, and comparing how the different cellular compartments are affected with chemical fixation. Ultrastructural analysis using serial section electron microscopy showed that chemically fixed tissue has a reduced volume: a significant loss of extracellular space and a decrease in the volume of neurites. However, the volume fraction of astrocytic elements increased following chemical fixation. In cryo-fixed tissue astrocytes showed a less intimate association with synapses, and their endfeet have a reduced coverage of blood vessels. Differences in membrane arrangements were also apparent at the synapse with a lower density of vesicles along the presynaptic membrane in chemically fixed synapses.

## Results

We first analysed how chemical fixation altered the volume of the neocortex by comparing fresh and chemically fixed brain sections cut through mouse primary somatosensory barrel cortex (*Figure 1A*, *Figure 1—figure supplement 1*). Coronal and tangential sections show the distinctive barrel pattern in layer IV, corresponding to the arrangement of whiskers on the mouse's muzzle, enabling us to estimate the total volume change. The chemical fixation protocol was a standard cardiac perfusion with buffered paraformaldehyde and glutaraldehyde, used widely as a preparation method for electron microscopy of brain tissue. Fresh sections were prepared from brains that had been removed rapidly from the skulls of decapitated mice and harvested in the same manner as for electrophysiological experiments.

Chemically fixed coronal sections showed enlarged ventricles and a clear reduction in total area compared to the fresh sections (*Figure 1A*). The cortical thickness was reduced by 16%, as measured from the pial surface to the start of the white matter (fresh $1.13 \pm 0.02$ mm, N = 6 mice; chemically fixed $0.95 \pm 0.07$ mm, N = 14 mice; p = 0.00004, unpaired two-tailed Student's t-test). Tangentially cut sections showed 18% shrinkage along the rostrocaudal axis, measured along the barrel rows: A1–A4, B1–B4, C1–C4 and D1–D4 (fresh $0.90 \pm 0.01$ mm, N = 3 mice; chemically fixed $0.74 \pm 0.03$ mm, N = 3 mice; p = 0.006, one way ANOVA). However, no shrinkage was found along the barrel arcs, on the mediolateral axis: A1–D1, A2–D2, A3–D3, A4–D4, A1–D4, A4–D1 (fresh $1.18 \pm 0.13$ mm, N = 3 mice; chemically fixed $1.19 \pm 0.11$ mm, N = 3 mice; p = 0.942, one way ANOVA). Taken together, these changes indicate that chemical fixation induced total volume shrinkage in the somatosensory neocortex of 30%.

Using serial section electron microscopy, we compared the neuropil from the chemically fixed brains with tissue samples that had been rapidly excised and cryo fixed using high-pressure freezing (*McDonald and Auer, 2006*) and resin embedded by freeze substitution (*Sosinsky et al., 2008*). The chemically and cryo-fixed tissue samples were similarly stained with heavy metals giving a suitable contrast to identify all the membranes and large macromolecular structures (*Figure 1C*). Cryo-fixed neuropil appeared qualitatively different from the chemically fixed tissue. Neuronal and glial processes were smooth and round, appearing to float in extracellular space. With chemical fixation, the neuropil showed markedly less extracellular space with membranes tightly apposed to each other, often with complex concave and convex shapes. Quantification of serial section electron micrographs revealed that the volume fraction of extracellular space in cryo-fixed neuropil was six times more than chemically fixed samples (*Figure 1D*; cryo fixation, $15.4 \pm 5.4\%$, N = 4 mice; chemical fixation, $2.47 \pm 1.5\%$, N = 4 mice; p = 0.003, one way ANOVA).

Further analysis of these volumes measured the contribution of the different cellular compartments. This showed that the astrocytic volume fraction in the cryofixed neuropil was half of the value for chemical fixation (*Figure 1E*, *Figure 1—figure supplement 1*; cryo fixed, $7.4 \pm 1.8\%$, N = 4 mice; chemical fixation, $14.4 \pm 3.3\%$, N = 4 mice; p = 0.01, one way ANOVA). In contrast, the volume fraction occupied by axons and dendrites was similar between the fixation conditions (*Figure 1—figure supplement 1*; cryo fixed $76.7 \pm 5.5\%$, N = 4 mice, chemically fixed $84.1 \pm 4.1\%$, N = 4 mice; p = 0.074, one way ANOVA). Chemical fixation therefore appears to induce an increase in the astrocytic volume fraction.

We next compared the structure of synapses under the two fixation conditions (*Figure 2*). Synapses were clearly visible in all material (*Figure 2A*), and measurements from serial images showed that chemically fixed neuropil had significantly higher synapse density than cryo fixed (*Figure 2B*; cryo fixed = $0.63 \pm 0.11$ µm$^{-3}$; chemically fixed = $0.87 \pm 0.15$ µm$^{-3}$, p = 0.042, one way ANOVA, N = 4 mice each group). The increased synapse density after chemical fixation is consistent with the overall

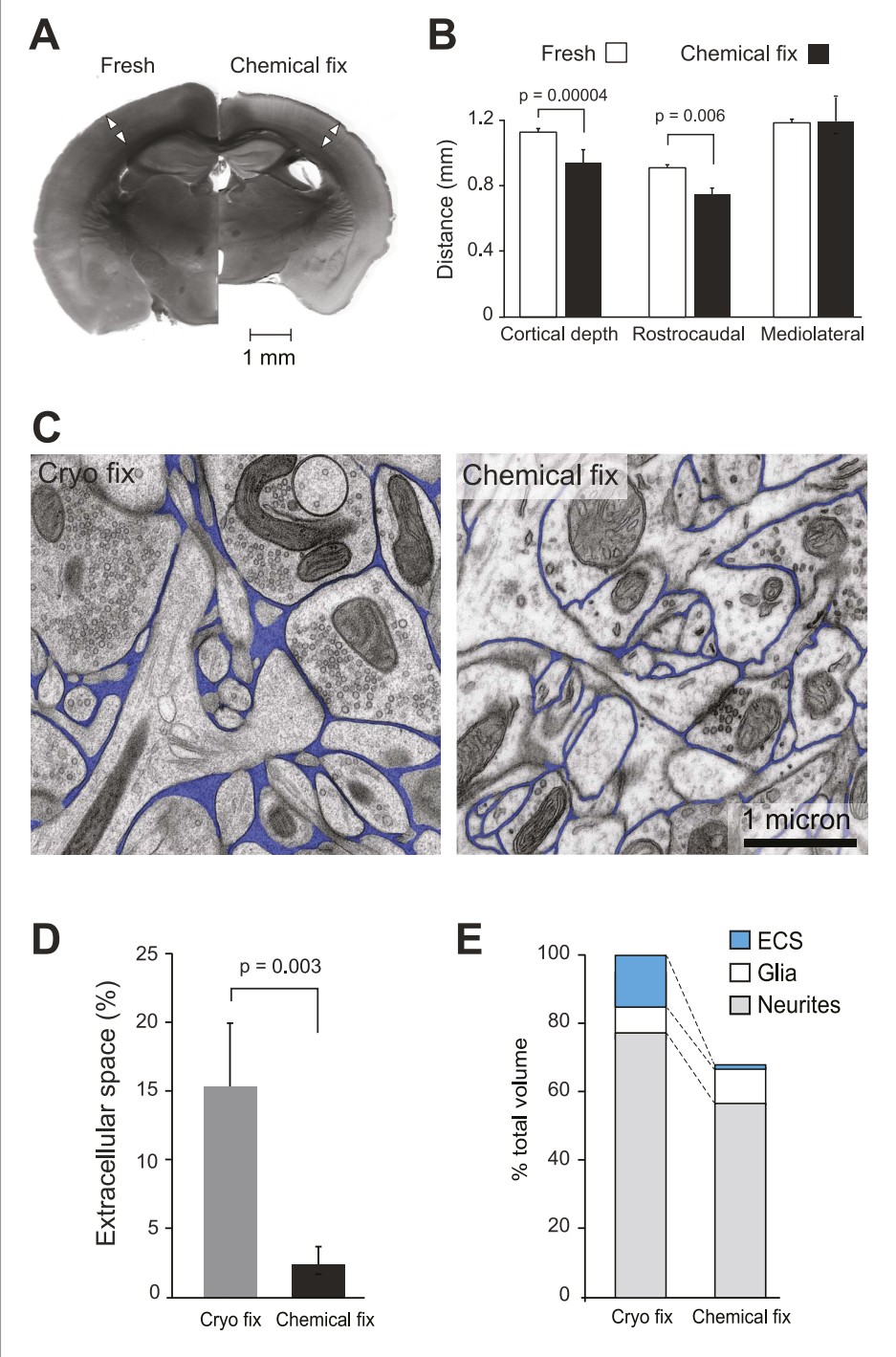

**Figure 1**. Chemical fixation reduces cortical volume and extracellular space. (**A**) Coronal sections of fresh (*left*) and chemically fixed (*right*) adult mouse brains. Double-headed arrows overlaying the somatosensory cortex of each section show the position at which the cortical thickness was measured. (**B**) Measurements of cortical thickness show a 16% reduction after chemical fixation (p = 0.00004, unpaired Student's t-test, *left*). Measurements across tangential sections show 18% shrinkage in the rostrocaudal axis (p = 0.006, one way ANOVA), but not in the mediolateral axis (p = 0.942, one way ANOVA, *right*). (**C**) TEM of cryo fixed (*left*) and chemically fixed (*right*) neuropil from the adult mouse cerebral cortex show reduction in the extracellular space (pseudo-coloured in blue) after chemical fixation. (**D**) Measurements of the volume fraction of extracellular space from serial section analysis showed a six-fold difference between the two fixation techniques (p = 0.003, one way ANOVA). (**E**) Measurements of volumes occupied by extracellular space, neurites, and glia, from serial section transmission electron microscopy sections
*Figure 1. continued on next page*

*Figure 1. Continued*

showed how the different compartments are altered by chemical fixation. Volume occupied by astrocytic processes was significantly increased after chemical fixation (p = 0.01, one way ANOVA). However, there was no change in the volume occupied by axons and dendrites (p = 0.074, one way ANOVA). As the volume of the cortex is reduced by 31% after chemical fixation, these percentages are shown in the bar chart in which the total volume of chemically fixed neuropil is 69% of the cryo-fixed value (100%).

The following source data and figure supplements are available for figure 1:

**Source data 1**. Data values and statistics underlying *Figure 1B, D, E*.

**Figure supplement 1**. Schematic scale model (upper image) representing the fresh somatosensory cortex (outer cube) and chemically fixed cortex (inner cube), showing the extent to which the two fixations change the volume of this brain region.

**Figure supplement 2**. Comparison between chemical fixation (left hand images; **A** and **C**) and cryo fixation (right hand images; **B** and **D**) of acute brain slices shows that both fixation conditions are able to reveal significant amounts of extracellular space.

volume shrinkage of the neocortex (*Figure 1*). Measuring the distance from the edge of spine synapses to the nearest cell membrane, showed a three times larger gap after cryo fixation compared to chemical fixation (*Figure 2B*; cryo fixation, 166 ± 18 nm, N = 3 mice; chemical fixation, 53 ± 5 nm, N = 3 mice; p = $2.7 \times 10^{-8}$, unpaired two-tailed Student's t-test). As astrocytes are present at many synapses, where they play a role in glutamate uptake, extracellular space homeostasis, and contribute to the regulation of synaptic transmission (*Ventura and Harris, 1999*; *Oliet et al., 2001*), we counted the proportion of spine synapses enveloped, or partially enveloped, by their processes (*Figure 2B*). Astrocytic processes at these types of synapses were significantly fewer in cryo-fixed cortex (cryo fixation, 34.0 ± 11.0%, N = 4 mice; chemical fixation, 62.4 ± 1.9%, N = 3 mice; p = 0.008, one way ANOVA). The numbers found in the chemically fixed neuropil are in good agreement with previous measurements (somatosensory cortex [*Genoud et al., 2006*]; hippocampus [*Harris and Stevens, 1989*]). Cryo-fixed tissue, in which there is a greater preservation of the extracellular space, therefore, reveals larger volumes around synaptic clefts suggesting that neurotransmitters can diffuse into large volumes of extracellular fluid before encountering other cell membranes.

The astrocytic elements in chemically fixed neuropil typically show less stained material in their cytoplasm compared to neurons. Their membranes appear to lie against the membranes of the surrounding axons and dendrites, giving them concave shapes, with a space-filling appearance, and large numbers of small processes squeezed between the neurites (*Figure 2C*). Reconstructions showed that cryo-fixed astrocytic processes stained similarly to axons and dendrites and were more rounded in appearance compared to astrocytes after chemical fixation (*Figure 2D*).

The apparent difference in appearance and arrangement of the astrocytic processes between the two fixation conditions was also investigated at the level of the blood capillaries (*Figure 3*), where their close association is suggested to play an important role in the regulation of solutes entering through the blood–brain barrier by almost completely surrounding the endothelial cells that form the vessel lumen (*Mathiisen et al., 2010*). By measuring the proportion of vessels that were surrounded by astrocytic endfeet (*Figure 3A,B*), we found significantly less coverage in cryo-fixed tissue (*Figure 3C*; percentage astrocytic coverage: cryo fixed 62.9 ± 15.0%, n = 11; chemical fixation 94.4 ± 6.0%, n = 69; p < 0.0001, unpaired two-tailed Student's t-test). Cryo-fixed tissue therefore reveals reduced astrocytic coverage of blood vessels suggesting that the abluminal surface of the endothelial cell has far greater direct access via the extracellular space to the neural elements within the brain than previously thought.

Synapses in chemically fixed tissue can be classified according to their morphology, and in the CNS they are typically categorized as either type 1 or type 2 (*Gray, 1959*). Type 2 synapses, later characterized as GABAergic, are distinguishable by their pre- and post-synaptic densities showing equal thickness (*Uchizono, 1965*; *Colonnier, 1968*), and their vesicles appearing flattened with darkened centres (*Figure 4A*, *Figure 4—figure supplement 1*). Measurements of the long and short diameters of synaptic vesicles at type 2 synapses in chemically fixed tissue indicate their ovoid

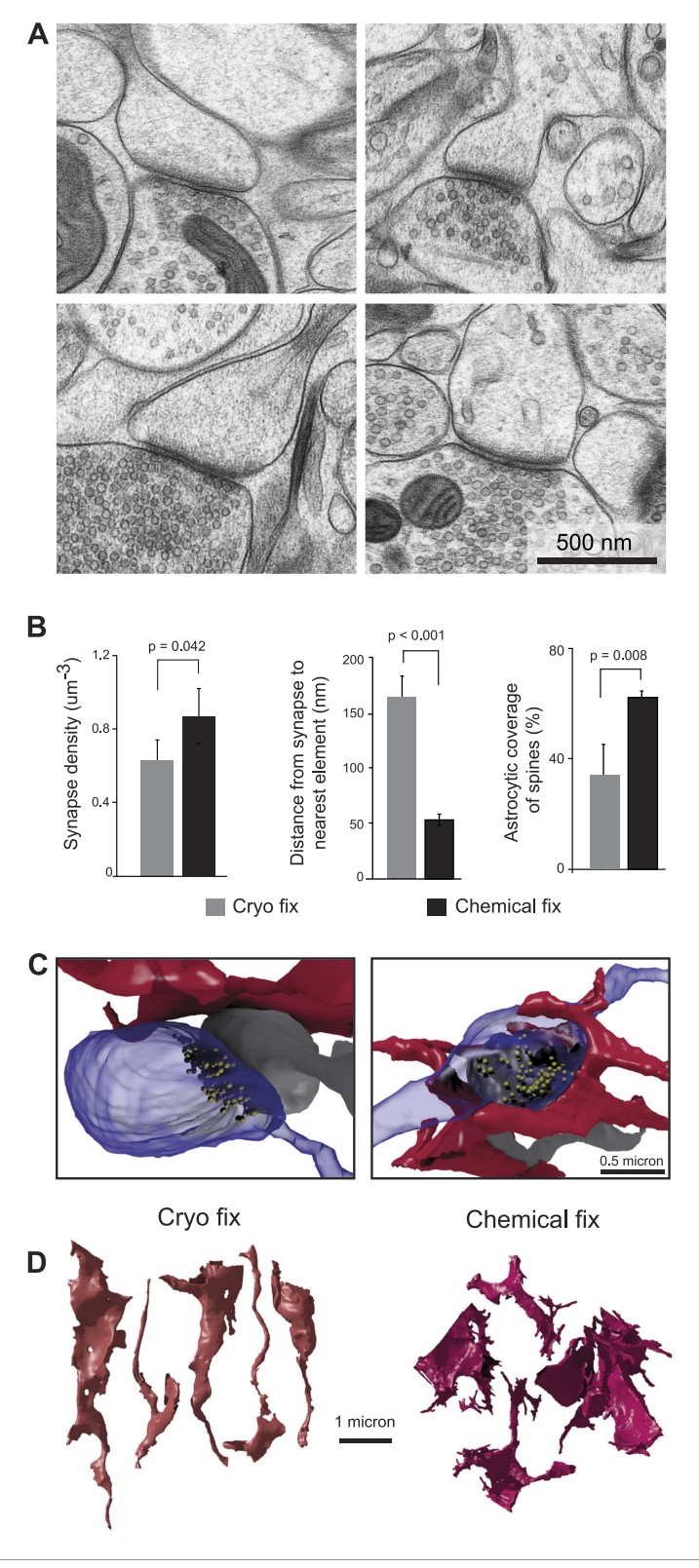

**Figure 2**. Cryo fixation reveals a larger peri-synaptic space and reduced astrocytic coverage. (**A**) Cryo-fixed neuropil shows synaptic contacts with large amounts of surrounding extracellular space. (**B**) Synaptic density measurements show the chemically fixed neuropil to have 38% more synapses (*left graph*, p < 0.05, one way ANOVA). Dendritic spine synapses (presumed glutamatergic) show greater distances between the edge of the contact zone and the
*Figure 2. continued on next page*

*Figure 2. Continued*

nearest membrane compared with chemical fixation (*middle graph*, p < 0.001, unpaired Student's t-test). Cryo-fixed synapses show less astrocytic coverage (*right graph*, p < 0.01; one way ANOVA). (**C**) Reconstructions from serial electron microscope images, of axonal boutons (blue) synapsing with dendritic spines (grey), show the astrocytic processes in the near vicinity (red). In the cryo-fixed synapse (*left*), the astrocytic process is not squeezed close to the synaptic contact (indicated with vesicles in yellow). In the chemically fixed example (*right*), the astrocyte tightly surrounds the synapse, where the vesicle-filled axonal bouton contacts the spine behind it. (**D**) Astrocytic processes reconstructed from serial FIBSEM images using the ilastik software (www.ilastik.org) show that chemically fixed astrocytic processes (*right*) have a more elaborate morphology with small processes extending from the flattened lamellae compared with the less complex structure of cryo-fixed astrocytes (*left*).

The following source data is available for figure 2:

**Source data 1**. Data values and statistics underlying *Figure 2B*.

shape (*Figure 4B*; long diameter 'y', 41.0 ± 5.4 nm; short diameter 'x', 28.2 ± 3.7 nm). The glutamatergic type 1 synapses in chemically fixed tissue, in contrast, have an obvious asymmetry with larger postsynaptic densities, and round vesicles with clear centres (*Figure 4—figure supplement 1*; diameter y, 40.0 ± 3.8 nm; diameter x, 39.1 ± 3.7 nm). In cryo-fixed neocortex all synapses, on spines and dendritic shafts showed similar symmetry, and vesicle diameters indicated them all as spherical (*Figure 4A,B*; diameter y, 38.7 ± 4.3 nm; diameter x, 38.5 ± 4.1 nm). This suggests that the chemical fixation causes the structural changes that allow this morphological distinction to be made. To check this, and verify that it was not an effect of the dehydration and embedding process, chemically fixed samples were high-pressure frozen and embedded at low temperature, by freeze substitution (*Sosinsky et al., 2008*). This material contained both asymmetric synapses (vesicle diameter y, 39.2 ± 4.0 nm; diameter x, 38.9 ± 4.8 nm) and synapses with symmetric pre- and post-synaptic densities, and flattened vesicles (*Figure 4*; diameter y, 41.6 ± 5.3 nm; diameter x, 27.7 ± 3.4 nm). The hallmarks differentiating glutamatergic and GABAergic synapses would therefore appear to be induced by chemical fixation.

We next compared the arrangement of synaptic vesicles in the two fixation conditions, measuring the distance of all vesicles within 150 nm of the presynaptic membrane, for synapses found on dendritic spines, larger than 0.2 microns and cut perpendicularly to the synaptic cleft (*Figure 5*). The average size of the synapses was the same in each group (*Figure 5—figure supplement 1*). The overall density of vesicles, within 150 nm of the presynaptic membrane, was the same in each group (cryo fixed, 37.9 ± 2.4 $\mu m^{-1}$, N = 3 mice; chemical fixed, 38.0 ± 2.6 $\mu m^{-1}$; N = 3 mice, p = 0.85; one way ANOVA). There were, however, clear differences in their spatial distribution (*Figure 5B*). The synaptic vesicle density within 30 nm of the presynaptic membrane was significantly increased in the cryo-fixed samples (*Figure 5B*; cryo fixed, 10.46 ± 0.88 $\mu m^{-1}$; chemical fixed, 2.99 ± 0.53 $\mu m^{-1}$; p < 0.001, unpaired Student's t-test). Between 30 and 60 nm this had decreased in cryo-fixed samples (*Figure 5B*; cryo fixed, 4.01 ± 0.56 $\mu m^{-1}$; chemical fixed, 8.02 ± 0.87 $\mu m^{-1}$; p < 0.001, unpaired Student's t-test). Beyond 60 nm there were no differences comparing cryo-fixed and chemically fixed synapses. This suggests that cryo fixation exposes two groups of vesicles; a group lying along the presynaptic membrane, and a second lying further back, away from the site of release.

## Discussion

In this study, we quantified differences in the ultrastructure of the adult mouse neocortex comparing standard chemical fixation with cryo fixation. Although chemical fixation is used widely and gives good ultrastructural preservation, it is also known to reduce the extracellular space and cause tissue shrinkage. Cryo-fixed tissue is likely to more closely resemble the native structure of the neocortex, and it is therefore of interest to compare cellular compartments and synaptic structures after cryo vs chemical fixation.

### Changes in the cellular compartments

The analysis of coronal slices at the level of the somatosensory cortex shows that chemical fixation reduced the tissue volume by 30%. This volume decrease would show an increase in synapse density of 43% (1/0.7). Our synaptic density measurements showed a 38% increase between the cryo-fixed

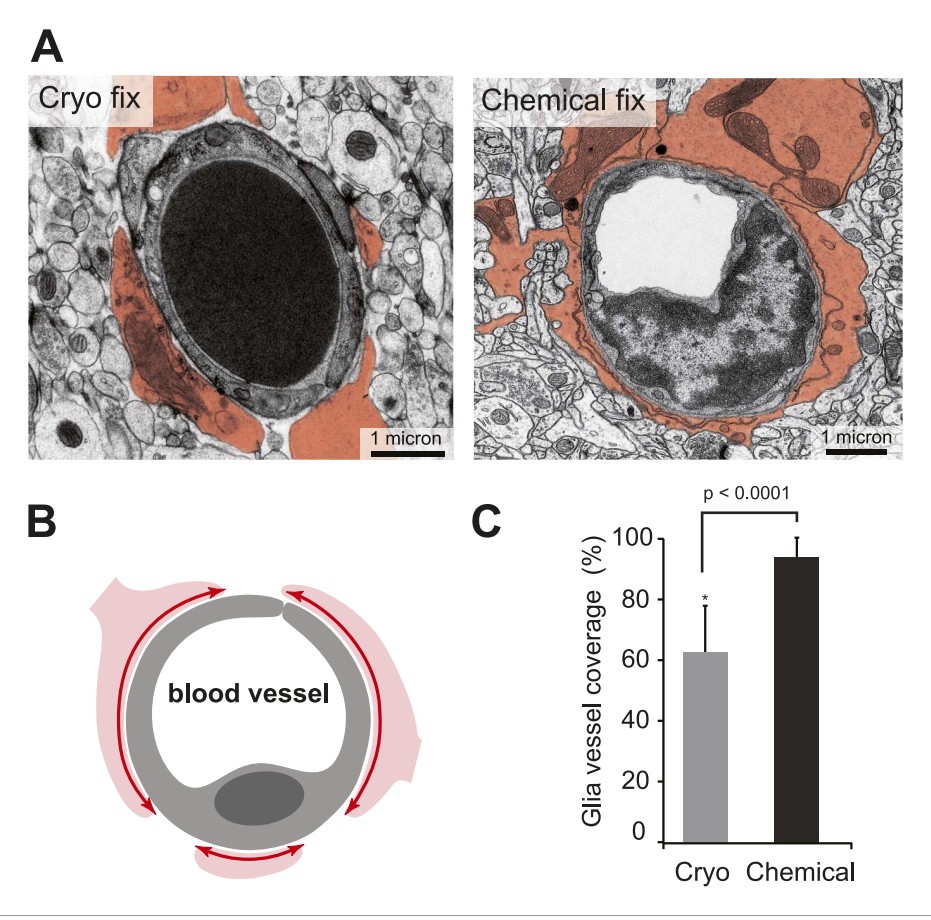

**Figure 3**. Cryo-fixed capillaries show less astrocytic coverage. (**A**) Electron micrographs of transversely sectioned capillaries show the astrocytic endfeet pseudo-coloured in orange. Cryo-fixed example shows a darkly stained erythrocyte within the vessel lumen. (**B**) Schematic diagram indicates the coverage measured. (**C**) Chemically fixed tissue contains capillaries with more glial coverage (p < 0.0001, n = 11 vessels cryo, n = 69 vessels perfused, unpaired Student's t-test).

The following source data is available for figure 3:

**Source data 1**. Data values and statistics underlying *Figure 3C*.

and chemically fixed tissue. The difference between these two values (43% vs 38%) might be accounted for by the fact that synapses were only counted in the neuropil, whereas the cortical volume measurements are from the whole tissue that includes cell bodies and blood vessels. We also cannot rule out changes to the tissue that may occur during the freeze substitution procedure as the acetone replaces the water at low temperature prior to any fixation with the osmium.

The fact that the chemical fixation process is relatively slow, involving the diffusion of aldehydes from blood vessels into the surrounding tissue, may also raise questions as to whether the short period of ischemia may cause a rapid assembly or disassembly of synaptic contacts. Sustained occlusion of cerebral vessels eventually leads to cell death and removal of synapses (*Kovalenko et al., 2006*). However, correlative in vivo light and electron microscopy studies would argue against this as they show that any anoxic changes initiated by the chemical fixation procedure do not result in the appearances or disappearances of any synaptic features such as dendritic spines or axonal boutons (*Trachtenberg et al., 2002*; *De Paola et al., 2006*; *Holtmaat et al., 2006*; *Knott et al., 2006*). Additionally, imaging the protein PSD95 in vivo also pinpoints all ultrastructurally identified synaptic connections, imaged with retrospective 3D electron microscopy (*Cane et al., 2014*), suggesting that it is unlikely that any synaptic formation or removal is initiated by the chemical fixation process.

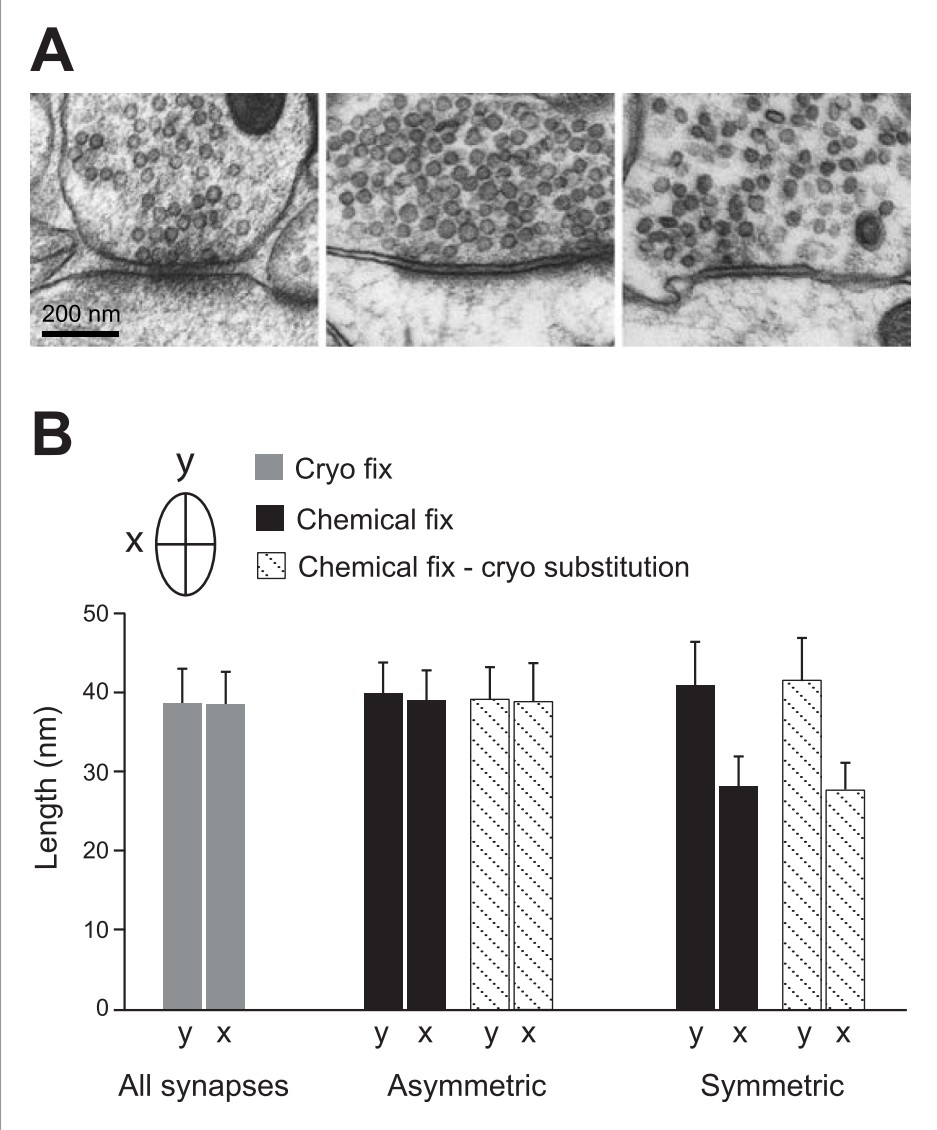

**Figure 4**. Vesicles of symmetric synapses are distorted by chemical fixation. (**A**) Cryo-fixed synapses on a dendritic shaft (*left image*) and on a dendritic spine (*middle image*) show similar rounded vesicles. A chemically fixed, high-pressure frozen and cryo-substituted (right hand image) synapse on a dendritic shaft, however, shows typical features of a symmetric (presumed GABAergic) synapse with ovoid vesicles. (**B**) Measurements of the short (x) and long (y) diameters of synaptic vesicles. Synapses in cryo-fixed tissue cannot be classified according to the symmetry of pre- and post-synaptic densities and all synaptic vesicles were round. Asymmetric synapses in chemically fixed tissue show similarly shaped vesicles, as do the vesicles at asymmetric synapses of chemically fixed tissue that is then high-pressure frozen and freeze substituted in resin. The symmetric synapses, seen in chemically fixed tissue, show vesicles with characteristic ovoid shapes irrespective of how they were resin embedded.

The following source data and figure supplement are available for figure 4:

**Source data 1**. Statistics underlying *Figure 4B*.

**Figure supplement 1**. Examples of glutamatergic synapses (**A**, **B**, **C**), situated on dendritic spines, with round clear vesicles; and presumed GABAergic synapses (**D, E, F**) on dendritic shafts showing flattened, dark vesicles.

To check that the high-pressure freezing itself does not cause a significant alteration of the tissue morphology, we fixed acute slices, prepared in the same manner as for electrophysiological recording, with the cryo fixation or by immersion in the chemical fixative. Both groups showed similar

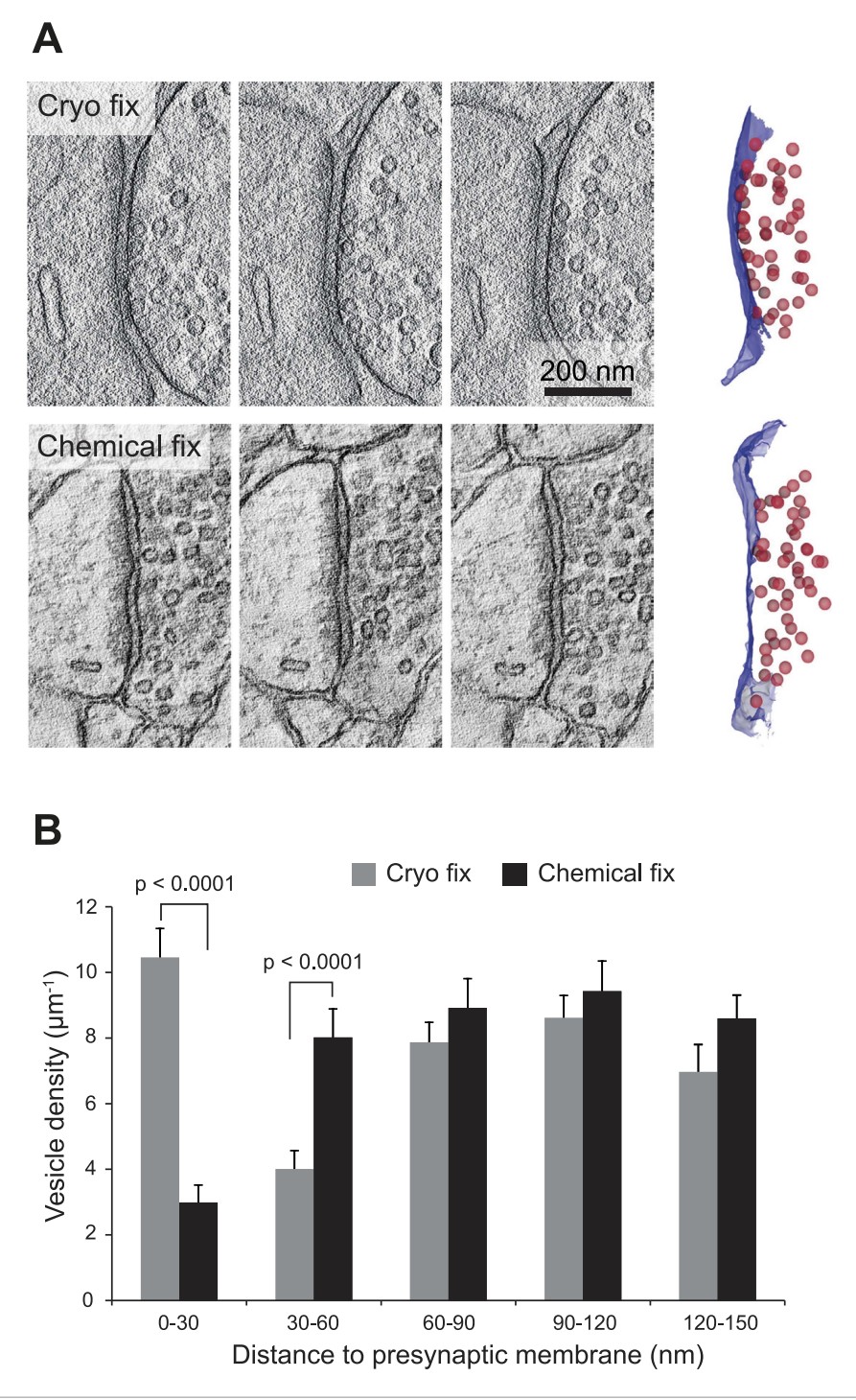

**Figure 5**. Cryo fixation preserves larger numbers of vesicles at the pre-synaptic membrane. (**A**) Electron tomography of a 200-nm thick section shows a cryo-fixed (*upper*) synapse with a large number of vesicles close to the presynaptic membrane in comparison with a similar chemically fixed synapse (*lower*). In each case three sample images are shown from complete tomographic series. Three-dimensional reconstructions of this region (right hand images) show all the vesicles (red) in relation to the presynaptic membrane (blue). (**B**) Measurements of the distance of vesicles from the presynaptic membrane show that more vesicles are arranged closer (0–30 nm) to the synapse after cryo fixation (p < 0.0001, unpaired Student's t-test).

*Figure 5. continued on next page*

*Figure 5. Continued*

The following source data and figure supplement are available for figure 5:

**Source data 1**. Data values and statistics underlying *Figure 5B*.
**Figure supplement 1**. Synapses found on dendritic spines were the same length.

tissue quality and levels of extracellular space, at least where ice crystal formation had not disrupted the ultrastructure (*Figure 1—figure supplement 2*).

In terms of how the different cellular compartments react to the chemical perfusion fixation, the earliest investigations were aware of the discrepancy between physiological and structural measurements. The 'watery look of the astrocyte' led to suggestions that if the physiological measurements of extracellular space were correct then 'methods of fixation here applied must be erroneous…' (*Schultz et al., 1957*). Observations like this and many others preceded cryo-fixation studies, first undertaken by Anton Van Harreveld. In Van Harreveld's experiments, exposed brain surfaces were instantaneously fixed, using cooled metal plates, to reveal a physiological level of extracellular space. However, Van Harreveld also found that delaying the process until after the onset of anoxia, preserved a neuropil similar to that seen with chemical fixation (*Van Harreveld et al., 1965*). This was the first illustration of how the extracellular space rapidly disappears once the heart has stopped. More recent live imaging of fluorescent diffusion markers showed a similar reduction (*Thorne and Nicholson, 2006*) suggesting that the process of cardiac perfusion for chemical fixation would induce a normal tissue response to anoxia resulting from the removal of the blood supply.

The reduction of extracellular space cannot account solely for the total tissue volume decrease seen after chemical fixation. Analyses to calculate the volume fractions of different cell components show that the neuronal part remains unchanged. However, considering the total tissue volume has shrunk by 30% the neuronal portion has therefore reduced similarly to maintain the same volume fraction. The doubling of the astrocytic volume fraction after chemical fixation could reflect how this cell type reacts in the aftermath of anoxia. Oxygen depletion elicits an energy reduction resulting in ions moving down their concentration gradients, with an increase in extracellular potassium, and wholesale depolarisation and spreading depression throughout the tissue (*Van Harreveld and Malhotra, 1967*; *Lutz, 1992*). The critical role that astrocytes play in buffering potassium in the extracellular space (*Kofuji and Newman, 2004*; *Binder et al., 2006*) and water movement during brain ischemia (*Manley et al., 2004*) point to these cells as playing the principal role in removing the extracellular space. The high concentrations of aquaporin transporters along their surfaces (*Benfenati and Ferroni, 2010*) suggest that astrocytes may be responsible for the removal of extracellular water ensuring that any changes in outside environment are rapidly neutralised. Cryo fixing brain tissue at different stages after the initiation of spreading depression, or ischemia, has shown similar astrocytic morphologies to those seen after chemical fixation (*Van Harreveld and Malhotra, 1967*).

The swelling of the astrocytic compartment, and removal of the extracellular space in response to chemical fixation, has important consequences for the interpretation of the ultrastructure. The perisynaptic region in chemically fixed tissue shows smaller spaces within which neurotransmitters can diffuse (*Van Harreveld et al., 1965*; *Ohno et al., 2007*) (*Figure 1*), with smaller distances between the edge of the synapse and surrounding elements. This region has variously been described at dendritic spines with a significant presence of astrocytes. In the neocortex, 60–70% of the bouton/spine interfaces are partially or completely surrounded with an astrocytic process (*Genoud et al., 2006*). A similar proportion was found in the hippocampus (*Harris and Stevens, 1989*). The larger perisynaptic space suggested by cryo fixation to exist in vivo would give neurotransmitters a greater opportunity to diffuse more rapidly out of the synaptic cleft, into the enlarged extracellular space where there would be a greater dilution, possibly leading to a slowing of their diffusion. The enlarged perisynaptic space could effectively act as a buffer zone, reducing extrasynaptic neurotransmitter concentrations, which could help isolate synapses, reducing the possibility that their activity would influence other extrasynaptic receptors, and minimising synaptic crosstalk. However, careful computational simulations (*Rusakov and Kullmann, 1998*; *Hrabe et al., 2004*) taking into account spatiotemporal dynamics, tortuosity, and binding/unbinding of neurotransmitters to various proteins (for example, glutamate transporters largely

located on astrocytic membranes) are essential to gain a more detailed understanding of neurotransmitter diffusion, and how this would be affected by the differences in ultrastructure we found after cryo fixation compared to standard chemically fixed tissue. The reduced synaptic density in the cryo-fixed tissue also places them further from each other, adding to their anatomical isolation. Cryo fixation therefore appears to suggest a reduction in the influence of volume transmission revealing an ultrastructure that might favour wiring transmission (*Kullmann, 2000*).

A more direct access through an enlarged extracellular space was apparent at the blood capillaries, another site where astrocytic processes have a significant presence. Here, their endfeet completely enclose blood capillaries after chemical fixation (*Mathiisen et al., 2010*) but cryo fixation reveals a partial coverage of about two thirds (*Figure 5*). The abluminal surface of the endothelial cell, therefore, is less insulated from the neuronal elements suggesting that solutes passing through the capillary wall from the blood have more direct access to the neurons. Conversely, substances released from neurons have greater access to blood vessels, where they might contribute to controlling blood flow (*Attwell et al., 2010*).

## Changes to the synapse

Other changes in the ultrastructure were seen at the synaptic connections. The classical morphology of inhibitory and excitatory synapses: their differences in symmetry and vesicle shapes that have proved useful for classifying types of connections in electron microscopy studies were not apparent in the cryo-fixed tissue. The flattened appearance of synaptic vesicles in GABAergic boutons therefore appears to be induced by the chemical fixation process.

Previous studies have shown that delaying chemical fixation by a few minutes alters synaptic structure. Asymmetric and symmetric contacts showed greater curvature and thicker postsynaptic densities (*Martone et al., 1999*; *Kovalenko et al., 2006*; *Tao-Cheng et al., 2007*). Pre-synaptically, vesicles were located further from the active zone. Images of the frog neuromuscular junction showed that chemical fixation, unlike freezing, could not prevent the fusion of vesicles to the presynaptic membrane (*Heuser et al., 1976*). More recently, the use of high-pressure freezing, simultaneous to synapse stimulation, has revealed how the vesicles fuse and collapse into the pre-synaptic membrane within 30 ms of stimulation (*Watanabe et al., 2013*). Analyses of vesicle arrangement in cryo-fixed CNS synapses in cultured slices (*Zhao et al., 2012*) or synaptosome preparations (*Fernández-Busnadiego et al., 2010*) have revealed a high concentration of vesicles clustered at the pre-synaptic membrane with the lowest concentration immediately behind this zone, 50–70 nm further back. However, after stimulation, the lowest concentration is seen at the pre-synaptic membrane steadily increasing further away (*Fernández-Busnadiego et al., 2010*). A similar picture is seen in our comparison between chemical and cryo-fixed cortical synapses (*Figure 3*). After cryo fixation, the greatest density of synaptic vesicles is at the pre-synaptic membrane, but after chemical fixation the vesicle density increases away from the pre-synaptic membrane suggesting that chemical fixation is unsuitable for capturing all docked vesicles.

Cryo fixation is able to reveal two groups of synaptic vesicles: one close to the synaptic membrane, and the other further back, approximately a vesicle width away. This peak in vesicle density close to the presynaptic release site has also been seen at the calyx of Held synapse (*Han et al., 2011*). Whether this heterogenous distribution represents different functional pools is unproven. However, a functional analysis of calyx synapses showed that disruption to the pre-synaptic protein machinery results in less vesicular release, together with a concomitant reduction in the number of vesicles seen close to the presynaptic membrane (*Han et al., 2011*). This would support the idea that vesicles against the presynaptic membrane are docked and associated with their release machinery (*Harlow et al., 2001*) where they can be depleted within a few milliseconds (*Felmy et al., 2003*). This arrangement of vesicles with cryo fixation could be revealing the readily releasable pool; those aligned tightly along the membrane. And the second grouping, further back from those in a docked position may, therefore, be those of the recycling and reserve pool, immobilized and clustered together within the body of the bouton by proteins such as actin or synapsin (*Hilfiker et al., 1999*; *Siksou et al., 2007*).

Cryo fixation, therefore, can provide a view of tissue ultrastructure that is closer to its natural state. This increases the relevance of morphological analyses by revealing an arrangement of cell membranes that more closely matches functional measurements. However, although these analyses highlight how the exquisite organization of the brain's cellular elements is acutely sensitive to chemical fixation, there

are difficulties in cryo preserving large volumes of brain tissue. The high water content of the CNS appears to limit the depth to which vitrification can occur (less than 10–20 microns in our hands at least) before ice crystal formation causes significant damage to the structure. Large-scale preservation for ultrastructural analysis will therefore continue to rely on chemical fixation approaches.

## Materials and methods

### Preparing fresh brain tissue for morphological measurements and cryo fixation

Adult mice (C57 BL/6, 7–10 weeks old) were decapitated, and the brain immediately removed. This was then immersed in ice-cold artificial cerebrospinal fluid (ACSF) composed of (in mM): 125 NaCl, 25 NaHCO$_3$, 1.25 NaH$_2$PO$_4$, 2.5 KCl, 0.1 CaCl$_2$, 3 MgCl$_2$, 25 glucose, 3 myo-inositol, 2 Na-pyruvate, 0.4 ascorbic acid, pH 7.4 which was bubbled with 95% O$_2$ and 5% CO$_2$. Slices were made with a vibratome (Leica Microsystems VT1200) at 150 microns thickness in the coronal or tangential plane. After cutting, each slice was then transferred immediately to a Petri dish containing the same medium and photographed with a stereo microscope (Leica Microsystems M205C).

For cryo fixation, immediately after decapitation, the brain was exposed using blunt surgical tweezers, and a piece of the somatosensory cortex cut from the brain using a razor blade. The tissue was then further sliced, and pieces, approximately 200-μm thick, were placed inside a 6 mm diameter aluminium sample holder with a 200-μm deep cavity. To ensure that no air bubbles became trapped inside the sample holder with the tissue, a small drop of 1-hexadecene was added. This was then high-pressure frozen using a Leica EM HPM100 (Leica Microsystems). This entire procedure was completed as rapidly as possible, in less than 90 s from the moment of decapitation. The frozen samples were then stored in liquid nitrogen until further processing.

### Preparing brain tissue with conventional cardiac perfusion of chemical fixatives

The animals were deeply anesthetized with pentobarbitone (0.05 mg g$^{-1}$) and perfused via the heart, using a perfusion pump, at a speed of 7 ml/min with 2.5% glutaradehyde, and 2% paraformaldehyde, in phosphate buffer (0.1 M, pH 7.4, 250–300 ml per animal). The tubing used for perfusion was back filled initially with 5 ml of 0.1 M PBS (pH 7.4) to help remove all blood from the circulatory system before the fixative entered. After perfusion, the animal was left for 1 hr and the brain was removed. 70-μm thick slices were vibratome sectioned in PBS (0.01 M, pH 7.4). These slices were then stained and embedded using previously described methods (*Knott et al., 2011*). Briefly, the slices were washed in cacodylate buffer (0.1 M, pH 7.4, 3 × 5 min), post fixed in 1% osmium tetroxide and 1.5% potassium ferrocyanide in cacodylate buffer (0.1 M, pH 7.4, 40 min). They were then stained with 1% osmium tetroxide in cacodylate buffer (0.1 M, pH 7.4) for 40 min, and then in 1% uranyl acetate for 40 min before being dehydrated in a graded alcohol series, 3 min each change, and embedded in Durcupan resin.

### Freeze substitution resin embedding of high-pressure frozen samples

Frozen tissue was stained, dehydrated, and embedded at low temperature using a low temperature-embedding device (AFS2; Leica Microsystems). Frozen samples were transferred to this device in liquid nitrogen and were then initially exposed to 0.1% tannic acid in acetone, for 24 hr at −90°C, followed by 12 hr in 2% osmium tetroxide in acetone at −90°C. The temperature was then raised over 4 days to −30°C, and then the liquid replaced with pure acetone and the temperature increased to −10°C over 16 hr. Finally, the tissue samples were mixed with increasing concentrations of resin over 8 hr whilst the temperature rose to 20°C. They were then added to 100% resin for 2 hr and then placed in silicon moulds for 24 hr at 65°C for the resin to harden.

### Electron microscopy

Cryo-fixed tissue that showed no artifacts of fixation, such as ice crystal damage, was selected by cutting semi-thin (0.5 mm thick) sections (2 × 2 mm) of the resin-embedded material and staining these with toluidine blue. With transmitted light microscopy, an area of well-fixed tissue could be identified and these regions were further trimmed and either used for TEM or FIBSEM.

### TEM imaging

Ribbons of serial sections (50–100 sections) were mounted onto single slot copper grids holding a polymer (pioloform) support film. Serial images were collected using a Tecnai Spirit (FEI Company) 120 kV transmission electron microscope operating at 80 kV, with an Eagle 4k × 4k CCD camera. Electron tomography imaging was performed on single, 200 nm thick sections using a Tecnai F20 (FEI Company) transmission electron microscope.

### FIBSEM imaging

Some of the material was imaged with FIBSEM (Zeiss NVision40) using the same preparation and imaging technique as described previously (*Knott et al., 2011*). Briefly, the trimmed blocks were mounted with conductive paint to aluminium stubs and gold coated (Cressington). This was then placed inside the microscope, and after ion milling to expose the required region, an imaging electron beam of 1.6 kV was used with a milling beam of 30 kV and 800 pA. Each milling and imaging cycle took approximately 90 s, with a pixel dwell time for the electron beam of 10 microseconds. Six hundred to a thousand images were collected by sequentially milling and imaging, with final image sizes of 2048 by 1536 pixels. Each pixel was 5 by 5 nm.

## Image analysis and morphometric measurements

All quantitative analysis of the serial images was carried out using the FIJI software (http://pacific.mpi-cbg.de/wiki/index.php/Fiji). Image alignments were performed using the stackreg plug-in.

Synapse density measurements were carried out using serial images through volumes of neuropil, containing no cell bodies or blood vessels. The images were displayed in the TrakEM2 software within FIJI (*Cardona et al., 2012*) and counted when they were completely contained within the displayed volume or were touching the left, top, and upper sides of the counting frame. Those touching the right, bottom, and lower sides were excluded. The section thickness was measured by using the mitochondria as cylindrical objects, and counting over how many sections they were present when lying parallel to the imaging plane (*Fiala and Harris, 2001*).

Measurements of volumes occupied by the different compartments were made by manually segmenting each of the different elements in the serial images using the TrakEM2 software. Area lists were made of the different compartments; neurites (axons and dendrites), astrocytes, and extracellular space, and drawn in each of the serial images. Distance measurements of cortical shrinkage, glial coverage of blood vessels, synapse and vesicle sizes, and vesicle and membrane separations were made using the same software.

Models of cellular elements shown in *Figure 3* were made using the interactive segmentation tool in the ilastik software (www.ilastik.org) from serial FIBSEM images (*Straehle et al., 2011*). These models were then imported into the 3D modelling software Blender (www.blender.org) for final composition and rendering.

## Acknowledgements

We thank Thorben Kroeger, Christoph Straehle, and Fred Hamprecht for help with using Ilastik software; Marco Cantoni and Davide Demurtas at the Centre of Electron Microscopy, EPFL, for help with FIBSEM imaging, and electron tomography. This work was funded by grants from EPFL-Stoicescu program (NK, CCHP, GK); Swiss National Science Foundation, SNSF (CCHP); Human Frontier Science Program, HFSP (CCHP); and European Research Council Advanced Grant, ERC (CCHP).

## Additional information

### Funding

| Funder | Author |
| --- | --- |
| Schweizerische Nationalfonds zur Förderung der Wissenschaftlichen Forschung | Carl CH Petersen |
| Human Frontier Science Program (HFSP) | Carl CH Petersen |

| Funder | Author |
| --- | --- |
| European Research Council (ERC) | Carl CH Petersen |

The funders had no role in study design, data collection and interpretation, or the decision to submit the work for publication.

## Author contributions

NK, Conception and design, Acquisition of data, Analysis and interpretation of data, Drafting or revising the article; CP, Conception and design, Drafting or revising the article; GK, Conception and design, Acquisition of data, Analysis and interpretation of data, Drafting or revising the article, Contributed unpublished essential data or reagents

## Author ORCIDs

Carl CH Petersen, http://orcid.org/0000-0003-3344-4495

## Ethics

Animal experimentation: All experiments were carried out in accordance with protocols approved by the Swiss Federal Veterinary Office (Canton of Vaud authorisation 1889.2).

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
