## [Decision Letter]

Thank you for sending your work entitled "Native ultrastructure of the mammalian neocortex revealed by cryo fixation" for consideration at *eLife*. Your article has been favorably evaluated by a Senior editor, a Reviewing editor, and two reviewers.

The following individuals responsible for the peer review of your submission have agreed to reveal their identity: Michael Häusser (Reviewing editor) and Tom Bartol (peer reviewer). A further reviewer remains anonymous.

The Reviewing editor and the reviewers discussed their comments before we reached this decision, and the Reviewing editor has assembled the following comments to help you prepare a revised submission.

The reviewers agreed that this study nicely highlights the changes that can occur when nervous tissue is chemically fixed by transcardial fixation. As an alternative, to circumvent tissue shrinkage and the potential redistribution of the extracellular space and cellular processes or synaptic vesicles, the authors used high-pressure freezing of the tissue. This method is advantageous since no aldehydes are applied that cause tissue shrinkage but is disadvantageous with respect to the study of larger tissue samples. Only small tissue blocks, such as brain slices, can be subjected to high-pressure freezing. When comparing cryofixed tissue with chemically fixed tissue the authors did indeed find considerable differences.

This is an important and timely analysis as new large-scale, high-throughput methods for 3D reconstructions of neuropil are becoming available. While the problem addressed in this manuscript has been a focus of discussion in the field for many years, the authors have managed to shed significant new light on this issue. Overall the manuscript is clearly written and the analysis and figures present the case very well. However, there was one major concern, described in detail below, which will require a comparison of aldehyde-fixed brain slices and high-pressure frozen slices.

The authors compared tissue fixed by transcardial perfusion of aldehyde solution with fresh tissue that was taken out of the brain and sectioned prior to high-pressure freezing. Although there is rapid sealing of cell membranes, previous light and EM studies have shown that there is swelling of the tissue following slice preparation. This is not surprising since thousands of dendritic and axonal processes were transected and damaged during slice preparation. In contrast, perfusion fixation leaves all elements in place and there is no transection of unfixed cellular processes. How can the authors exclude that the changes they observed did not derive – at least in part – from the damage to fresh tissue during slice preparation? I wonder why the authors did not compare brain slices that were chemically fixed after slice preparation with those that were subjected to high-pressure freezing. There is in fact evidence from recent literature that the extracellular space is enlarged in acute hippocampal slices subjected to high-pressure freezing when compared to high-pressure frozen slice cultures. In contrast to acute slices, there is reorganization of the tissue and removal of debris in slice cultures incubated for several days or even weeks (Studer et al., 2014) Capture of activity-induced ultrastructural changes at synapses by high-pressure freezing of brain tissue. Nature Protoc 9:1480-1495). In conclusion, the authors should strengthen their study by comparing acute brain slices that were chemically fixed after slice preparation with acute slices subjected to high-pressure freezing.

---

## [Author Response]

*The authors compared tissue fixed by transcardial perfusion of aldehyde solution with fresh tissue that was taken out of the brain and sectioned prior to high-pressure freezing. Although there is rapid sealing of cell membranes, previous light and EM studies have shown that there is swelling of the tissue following slice preparation. This is not surprising since thousands of dendritic and axonal processes were transected and damaged during slice preparation. In contrast, perfusion fixation leaves all elements in place and there is no transection of unfixed cellular processes. How can the authors exclude that the changes they observed did not derive – at least in part – from the damage to fresh tissue during slice preparation? I wonder why the authors did not compare brain slices that were chemically fixed after slice preparation with those that were subjected to high-pressure freezing. There is in fact evidence from recent literature that the extracellular space is enlarged in acute hippocampal slices subjected to high-pressure freezing when compared to high-pressure frozen slice cultures. In contrast to acute slices, there is reorganization of the tissue and removal of debris in slice cultures incubated for several days or even weeks (Studer et al., 2014) Capture of activity-induced ultrastructural changes at synapses by high-pressure freezing of brain tissue. Nature Protoc 9:1480-1495). In conclusion, the authors should strengthen their study by comparing acute brain slices that were chemically fixed after slice preparation with acute slices subjected to high-pressure freezing*.

The reviewers are correct to point out that we do not know how closely the cryo fixed tissue resembles the native brain structure. We have therefore changed the Title of the manuscript and the text.

However, we think there is good reason to think that cryo fixation reveals ultrastructure that more closely resembles native brain structure because:

1) Cryo fixed brain tissue shows an extracellular space similar to that measured in vivo.

2) Cryo fixed brain tissue shows synaptic vesicles clustered near the presynaptic membrane.

Across the biological samples where it has been applied, there is general agreement that cryo fixation preserves ultrastructure in a near-physiological state, and we think the same is likely to be true under our experimental conditions investigating mouse neocortex.

The reviewers suggests that there might be ”swelling of the tissue following slice preparation”. Our measurements of cryo fixed tissue do not indicate swelling after cryo fixation, but rather suggest that astrocytes swell during chemical fixation. We would like to point out that we did not use the normal protocol for preparing acute slices, as detailed in Studer et al. (2014), and in being able to measure the same amount of extracellular space (ECS) as physiological studies, we think it is likely that our rapid extraction method, prior to freezing, was fast enough to limit the possible changes.

The reviewers suggest that we examine differences in chemical versus cryo fixation of acute brain slices, prepared in the same way as for electrophysiological experiments. This would check whether the changes we are seeing are not due to the influence of the different fixation methods themselves rather than the physiological state of the tissue.

We have carried out these experiments using the standard approach for slice preparations, as described by Studer et al., although we used 6 week old animals, and not ‘pups’ (Studer et al., 2014).

These slices were either dropped into chemical fixative or cryo fixed, after they had first been allowed to recover in medium for 40 min at 30°C. This warming is common practice in the preparation of acute slices ready for electrophysiology experiments. Electron microscopy shows that irrespective the fixation method, both preparations show large amounts of ECS (see Figure 1—figure supplement 2). Therefore, this would suggest that the removal of ECS that we see in the perfusion fixed material is not due to the fixation method itself, but rather its physiological state. If we chemically fix a slice prior to the recovery period, immediately after vibratome slicing (Figure 6; after step 19, Studer et al., 2014) the morphology is very similar to what we find after the perfusion fixation with a reduced extracellular space. It is also the same when cryo fixation is performed 5 min after the heart has stopped, and the same as described by Van Harreveld and Khattab (1967) after spreading depression.Author response image 1.Electron micrograph showing the neuropil from an acute slice chemically fixed immediately after it was vibratome sectioned. The image is taken 50 micons from the slice surface.**DOI:**
http://dx.doi.org/10.7554/eLife.05793.019

The amount of ECS that we see in these acute ‘recovered’ slices appears to be far greater than ever seen in freshly extracted tissue. This observation though comes from only looking within the first 10 microns of the cryo fixed tissue (Figure 1—figure supplement 2). We were not able to find well-preserved tissue deeper into the slice without considerable ice crystal damage (Figure 1—figure supplement 2). We think that this is because of the excessive amounts of watery solution that surrounds the cellular elements making the tissue difficult to vitrify to any significant depth.

As for carrying out a detailed morphological analysis of acute slices that have either been chemical or cryo fixed. It is clear that we are not able to preserve slices to any significant depth with cryo fixation. We must also recognise that these slices are bathed in an artificial CSF that does not exactly replicate in vivo conditions. Therefore, our analysis would only show the fixation differences within a severely disrupted region of brain tissue in an artificial environment.

The tests we show here (Figure 1—figure supplement 2) suggest that chemical or cryo fixation can reveal similar morphologies, but they also show that the levels of ECS can be influenced by the medium in which they are placed. Here, we find a level that appears to be excessively high.

The goal of our study was to compare the widely recognised standard fixation approach for brain tissue; namely chemical fixation via cardiac perfusion, with a rapid cryo fixation method of the fresh tissue. Our reasoning was that all large scale connectomics analyses of mammalian central nervous systems use chemical fixation, but to date there is little consideration for the natural state of the ultrastructure. We would like to draw the attention of the reviewers also to the recent paper by Mikula and Denk (Nature Methods, 2015), that illustrates why perfusion fixation of the brain is an important process for preserving the neural circuits of connectomics. It is therefore important to understand what changes occur to brain ultrastructure during chemical perfusion fixation, and we think our study makes important advances towards this goal.